# Assessing the Accuracy of a Three-Year High-Resolution Mesoscale Wind Farm Wake Simulation with Lidar and Satellite Radar Data

Alexandros Palatos-Plexidas<sup>1, 2</sup>, Simone Gremmo<sup>1</sup>, Jeroen van Beeck<sup>1</sup>, Lesley De Cruz<sup>2, 3</sup>, and Wim Munters<sup>1</sup>

**Correspondence:** Alexandros Palatos-Plexidas (alexandros.palatos-plexidas@vki.ac.be)

Abstract. The rapid expansion of wind farm installation in the North Sea results in an increased need for understanding their influence on the local atmosphere, as well as the interactions between them. Wind farm operation and power production are affected by wakes produced both within the wind farm and by upwind wind turbines. To accurately estimate wind power production, it is essential to quantify the effects behind these extended wind speed deficits using mesoscale atmospheric modelling. This study presents a three-year-long mesoscale analysis using the Weather Research and Forecasting (WRF) model at a horizontal resolution of 1 kilometer. The simulations are evaluated against lidars located in the Southern Bight of the North Sea in the vicinity of the large Belgian-Dutch offshore wind farm cluster, illustrating that the model performs adequately. Coupling the mesoscale atmospheric model with the Fitch wind farm parameterization (WFP) scheme significantly improves simulation accuracy, particularly in regions frequently affected by wake effects. An analysis of the model performance under different atmospheric boundary layer (ABL) stratification conditions shows that the model performs better under less extreme stability cases, while a complementary evaluation of upstream, intra-farm, and downstream wake characteristics further highlights the benefits of using the Fitch WFP scheme in WRF. In addition, synthetic aperture radar images are compared to model outputs for specific wake events, indicating that the wind farm parameterization scheme effectively captures wake structures at the analyzed timestamps.

#### 15 1 Introduction

In recent years, wind energy has become one of the most promising electricity generation technologies for a sustainable energy system. Especially offshore, wind farms are growing in size and number, resulting in gigawatt-scale clusters being installed in each other's vicinity. Therefore, it is important to quantify the effects of wind farms both in terms of how their interaction impacts the operation and energy production, as well as their local influence on the atmosphere. These effects vary based on the predominant atmospheric conditions and wind characteristics.

<sup>&</sup>lt;sup>1</sup>von Karman Institute for Fluid Dynamics, Environmental and Applied Fluid Dynamics, Waterloosesteenweg 72, 1640 Sint-Genesius-Rode, Belgium

<sup>&</sup>lt;sup>2</sup>Vrije Universiteit Brussel, Electronics and Informatics Department, Pleinlaan 9, 1050 Brussels, Belgium

<sup>&</sup>lt;sup>3</sup>Royal Meteorological Institute of Belgium, Observations and Research scientific services, Ringlaan 3, 1180 Brussels, Belgium

https://doi.org/10.5194/wes-2025-202 Preprint. Discussion started: 17 October 2025 © Author(s) 2025. CC BY 4.0 License.

30

50

One of the most pertinent tasks is the understanding and quantification of wake effects produced by wind farms. Wakes exert a substantial influence on downwind wind turbines, as they lead to decreased wind speeds and increased turbulence that affect both the power production and the fatigue of the wind turbine (Stanley et al., 2022; Jézéquel et al., 2024). Wakes are the dominant driver for power losses in offshore wind farms, reducing power extraction in the order of 10% to 20% compared to lone-standing turbines (Lee and Fields, 2021). Furthermore, especially for large wind farms, individual turbine wakes merge into wind farm wakes that have been shown to persist several tens of kilometers downstream of the wind farm, depending on the atmospheric conditions (Ali et al., 2023). As a result, offshore wind farms are affected by current and future neighboring farms, and reducing uncertainties on these interactions is becoming increasingly important for the efficient development of offshore wind energy.

High-fidelity simulations like large-eddy simulations (LES) can provide useful insights on the interaction between wind turbines and the atmospheric boundary layer (ABL), with different approaches for wind turbine representation being developed (Sorensen and Shen, 2002; Stevens et al., 2018; Stipa et al., 2024). However, computational cost for high-resolution LES remains very high, rendering them impractical for simulating large wind farm clusters over long periods. Furthermore, performing LES under real atmospheric conditions increases the computational cost further as proper coupling with a mesoscale model is required (Haupt et al., 2023; Muñoz-Esparza et al., 2014; Kale et al., 2023). As a result, mesoscale numerical weather prediction (NWP) models continue to be the most cost-effective state-of-the-art tool to assess atmospheric conditions spanning a wide geographic area and a prolonged period. NWP models typically operate on horizontal grid spacings of 1 to 3 km, necessitating the use of different parametrization schemes to capture the effect of subgrid-scale physics such as convection, hydrometeor microphysics, radiation, surface interactions, and turbulence. Over the past years, many NWP models have also included parametrization schemes to account for the influence of the wind turbines in the ABL. An extensive description of these wind farm parametrization (WFP) schemes in NWP models can be found in the work of Fischereit et al. (2022a). In particular, although several alternatives have been developed (Volker et al., 2015; Ma et al., 2022a), the most widely used WFP scheme in NWP is the method proposed by Fitch et al. (2012). Even though the scheme has significant limitations, e.g. on the turbulence kinetic energy (TKE) representation (Archer et al., 2020), the freestream wind estimation (Vollmer et al., 2024), and the inclusion of subgrid-scale wakes (Ma et al., 2022b), the original implementation has been shown to agree reasonably well with high-fidelity simulations (Archer et al., 2020; Peña et al., 2022; García-Santiago et al., 2024; Fischereit et al., 2022b), field measurements (Larsén and Fischereit, 2021; Ali et al., 2023; Fischereit et al., 2024), and turbine production data (Lee and Lundquist, 2017; Santoni et al., 2020; Sengers et al., 2025). Nevertheless, further validation studies over extended time periods are needed to reduce uncertainties in quantifying wind farm wakes and the resulting inter-farm interactions (Fischereit et al., 2022a).

The vast majority of WFP studies utilize the Weather Research and Forecast (WRF) model (Skamarock et al., 2019). WRF is an open source and widely used NWP model able to simulate complex atmospheric phenomena and extreme events (Larsén et al., 2019; Müller et al., 2024; Shenoy et al., 2021). It has been used in a variety of studies that estimate the influence of wind farm effects (see, e.g., García-Santiago et al., 2023; Pryor et al., 2019, 2020). More specifically, studies estimating the wind farm effects under different ABL stability conditions (Rosencrans et al., 2024; Palatos-Plexidas et al., 2024) have been

https://doi.org/10.5194/wes-2025-202

Preprint. Discussion started: 17 October 2025

performed. In addition, specific events like extreme winds (Pryor and Barthelmie, 2021), storms (Ivanova et al., 2025), and low-level jets are well reproduced by WRF at different locations across the globe (Vanderwende et al., 2015; Quint et al., 2025). Although there are a few studies that simulate the wind farm effects using mesoscale models at 1 km or sub-kilometer scales under real atmospheric conditions (Gomez et al., 2024), long-term high-resolution mesoscale simulations and validations remain largely absent from the literature.

The current work presents an assessment of model performance and analysis of wake effects from a three-year long high-resolution (1 km) WRF simulation of the Southern Bight of the North Sea, with a specific focus on the 3.7 GW Belgian-Dutch wind farm cluster. This cluster consists of several high-density wind farms installed next to each other, making it the largest offshore cluster operational today, hence resulting in a prime test case for studying wind farm wakes. Extending the analysis over a three-year period enables a comprehensive characterization of seasonal patterns and temporal variability.

The study is driven by two primary objectives. The first objective is to validate the simulation dataset using lidar measurements in the vicinity of the wind farm cluster, with particular attention to model performance under varying ABL stability conditions. The analyzed lidar probes are strategically located and enable to evaluate the wind fields associated with varying wind directions inside and outside the cluster. The second objective is to evaluate model performance at selected timestamps by comparing simulation wind speeds with synthetic aperture radar (SAR) data. The assessment is conducted both qualitatively—through analysis of the full two-dimensional wake structures—and quantitatively, by examining the transect connecting the analyzed lidar locations. This approach enables a detailed evaluation of wind speed deficits and model accuracy relative to SAR imagery and available lidar measurements.

The remainder of the study is structured as follows. Section 2 outlines the methodology, including the WRF model setup, definitions of performance metrics, and classification of ABL stability. Next, Section 3 highlights the simulation results, including the overall atmospheric conditions and model performance assessment with lidar data under varying conditions of stability, seasonality, and wake loss intensity. Finally, Section 4 summarizes the main findings of this study.

## 2 Methodology

This section outlines the methodology used to create and analyze the high-resolution dataset. First, the configuration of the WRF model is described. Next, the definition and the classification of atmospheric stability regimes is introduced. Finally, the lidar observations and the SAR images used for model performance assessment are presented, followed by the evaluation metrics that are applied in this study.

# 2.1 WRF model setup

Simulations were performed with version 4.5.2 of the WRF model (Skamarock et al., 2019) over a period of three years: 2021, 2022, and 2023. The initial and boundary conditions are derived from 30 km spatial resolution ERA5 hourly reanalysis data (Hersbach et al., 2020). Three nested square domains are employed with grid resolutions of 9, 3, and 1 km, respectively, as depicted in Figure 1. In the vertical direction, 80 levels were used up to 50 hPa, while the first 30 levels are densely distributed

starting from approximately 5 meters up to 320 meters height, with an average vertical resolution of 11 m. Adaptive time stepping is used with a target Courant–Friedrichs–Lewy number of 0.84 and 0.6 for the horizontal and vertical directions respectively. In addition, vertical velocity damping is enabled to ensure robustness and stability during the year-long runs using a Rayleigh damping layer with a depth equal to 5000 meters from the model top and a damping coefficient of 0.15 ms<sup>-2</sup>. Results of the high-resolution domain d03 are processed every 10 minutes for comparison to lidar data. The model physics configuration is based on the New European Wind Atlas (NEWA) setup (Hahmann et al., 2020; Dörenkämper et al., 2020) and is presented in Table 1. The cumulus scheme is activated only at the outermost domain of 9 km resolution. Regarding the planetary boundary layer (PBL) turbulence modelling, the MYNN (Nakanishi and Niino, 2006) scheme is used, as it is the only scheme available in WRF v4.5.2 capable of advecting the TKE produced by the WFP as explained below. Note that, rather than performing a specific sensitivity study to optimize model performance for the location of interest, a single WRF configuration, close to the well-known NEWA setup, is chosen in order to evaluate typical out-of-the-box performance of the model against measurements.

**Table 1.** WRF physics parameterization schemes selected for this study, along with their associated references.

| Parameterization               | WRF Scheme     | Reference                  |
|--------------------------------|----------------|----------------------------|
| Planetary Boundary Layer (PBL) | MYNN 2.5 level | Nakanishi and Niino (2006) |
| Surface Layer Model            | ETA Similarity | Monin and Obukhov (1954)   |
| Land Surface Model             | NOAH LSM       | Mukul Tewari et al. (2004) |
| Cumulus                        | Kain-Fritsch   | Kain (2004)                |
| Microphysics                   | WSM5           | Hong et al. (2004)         |
| Radiation                      | RRTMG          | Iacono et al. (2008)       |

100

#### 2.1.1 Wind Farm Parametrization

For this study, the WFP scheme proposed by Fitch et al. (2012) is used. The influence of turbines on the atmosphere is described by imposing a momentum sink in the mean flow, which enables the conversion of a fraction of the kinetic energy (KE) into electricity, while the remaining KE is transformed into TKE. The WRF model is executed with and without wind farm parameterization activated (hereafter WF and NWF, respectively) within a single model run, using two identical d03 domains, to isolate wake effects. The study focuses on the investigation of the Belgian-Dutch cluster, however, nearby offshore wind farms in the Southern Bight of the North Sea, i.e. in the United Kingdom and in the Netherlands, are included in the simulations. The locations of all the wind turbines included in the simulations are obtained from the study of Hoeser et al. (2022). The wind farms are depicted in Figure 1(b) and they are exclusively simulated on the highest resolution domain. The principal equations for describing the wind farms influence on the mean flow in a discrete Cartesian computational cell i, j, k are defined as

120

**Figure 1.** (a): Map of the three nested domains used for the WRF simulations, with a horizontal resolution of 9 km (D01, blue box), 3 km (D02, black box) and 1 km (D03, red box). (b): Zoom on the high-resolution innermost domain (D03), including the simulated wind turbines (black dots).

$$\frac{\partial |\mathbf{V}|_{ijk}}{\partial t} = -\frac{N_t^{ij} C_T \left( |\mathbf{V}|_{ijk} \right) |\mathbf{V}|_{ijk}^2 A_{ijk}}{2(z_{k+1} - z_k)},\tag{1}$$

$$\frac{\partial P_{ijk}}{\partial t} = \frac{N_t^{ij} C_P \left( |\mathbf{V}|_{ijk} \right) |\mathbf{V}|_{ijk}^3 A_{ijk}}{2(z_{k+1} - z_k)},\tag{2}$$

$$\frac{\partial TKE_{ijk}}{\partial t} = \frac{N_t^{ij}C_{TKE}\left(|\mathbf{V}|_{ijk}\right)|\mathbf{V}|_{ijk}^3 A_{ijk}}{2(z_{k+1} - z_k)}.$$
(3)

These equations describe the change in horizontal wind speed |V|, the wind power production P, and the turbulence enhancement TKE due to wind turbines, where  $z_k$  defines the height at model level k, and  $A_{ijk}$  is the cross-sectional rotor area of one wind turbine between the height levels k and k+1. Furthermore,  $N_t^{ij}$  is the amount of turbines located within cell (i,j). The thrust coefficient  $C_T$  and power coefficient  $C_T$  depend on the wind turbine type and are obtained from a mix of public and confidential data. The TKE coefficient is defined as  $C_{TKE} = \alpha(C_T - C_T)$ , with  $\alpha = 1$  in the original Fitch implementation, driven by the idea that momentum lost by the mean flow which is not converted to electrical power is transformed into turbulence. However, it was found that this can lead to a significant overestimation of the TKE compared to high-fidelity simulations (Archer et al., 2020). Though there is still significant debate on the appropriate value for  $\alpha$  (Larsén and Fischereit, 2021; García-Santiago et al., 2024), this study uses the current default value in WRF, i.e.,  $\alpha = 0.25$ . Turbulence advection is

© Author(s) 2025. CC BY 4.0 License.

130

135

140

activated in the high-resolution domains. The wind speed deficit due to the wind turbine influence is defined as the difference between the unperturbed and the wind-farm-affected wind speed, i.e.,  $\Delta U_{\rm wake} = U_{\rm NWF} - U_{\rm WF}$ .

# 2.2 Atmospheric Boundary Layer stability

The stability conditions in the lower layers of the ABL can be assessed by analyzing temperature, wind speed, and surface heat fluxes (Stull, 2015). These variables are closely associated with atmospheric turbulence and the resulting wake mixing. In this work, the atmospheric stability is characterized by the value of Monin-Obukhov (MO) length L (Monin and Obukhov, 1954), as retrieved from the PBL scheme in the inverse MO length (RMOL) variable. The MO length compares how shear and buoyancy contribute to the production and dissipation of TKE, i.e.,

$$L = -\frac{u_*^3 \overline{\theta}_v}{\kappa g \left( \overline{w'\theta'_v} \right)_c},\tag{4}$$

where  $u_*$  is the friction velocity,  $\overline{\theta}_v$  is the mean virtual potential temperature at the first model level above the surface (approximately at 5 m),  $\kappa = 0.41$  is the von Kármán constant, g is the gravitational acceleration and  $(\overline{w'\theta'_v})_s$  is the surface virtual potential heat flux. Stable conditions are present when L > 0, when shear production and buoyancy destruction counteract leading to limited turbulence and intermittency. On the contrary, when L < 0, buoyancy enhances turbulence, resulting in stronger vertical motion and mixing. In neutral conditions, when  $|L| \to \infty$ , buoyancy has a negligible effect and turbulence is driven solely by shear. Besides the main stable, unstable, and neutral conditions, four additional ABL stratification categories as defined in Table 2 are considered in the analysis, seeking to deliver a more comprehensive representation of the stability conditions. This approach is similar to the work of Gryning et al. (Gryning et al., 2007), and has been implemented in the studies of Olsen et al. (Olsen et al., 2022) and Palatos-Plexidas et al. (Palatos-Plexidas et al., 2024).

**Table 2.** Atmospheric stability classes. Monin-Obukhov values are calculated in WRF using Equation (4) and seven classes are selected (Gryning et al., 2007).

| Stability classes | Monin-Obukhov length values (m) |  |  |
|-------------------|---------------------------------|--|--|
| Very stable       | $0 < L \le 50$                  |  |  |
| Stable            | $50 < L \le 200$                |  |  |
| Stable-Neutral    | $200 < L \leq 500$              |  |  |
| Neutral           | L  > 500                        |  |  |
| Unstable-Neutral  | $-500 

#### 2.3 Offshore Observations

There are two observations sources that are used in this work and they are presented in this section. Initially, lidar profile measurements are employed to assess model performance throughout the entire analysis period. Subsequently, Synthetic Aperture Radar (SAR) data from the Southern Bight of the North Sea are utilized to compare and validate the model-assessed wakes at specific timestamps.

## **2.3.1** Lidars


Model performance is evaluated against ten-minute average wind speed and wind direction measurements from four vertical profiling lidars located in the Southern Bight of the North Sea, close to the Belgian-Dutch wind farms cluster. As discussed below, considering the wind rose at this site, the locations of the lidars (see Figure 2) provide a unique opportunity for assessing wake effects. Details on their location and the operation period during the simulation time window are described in Table 3. The following lidars are considered:

- WHi: The Westhinder (WHi) lidar is a ZX 300M installed on the Westhinder (MP7) survey platform located approximately 40 km southwest of the Belgian-Dutch cluster, close to the French border in Belgium. The lidar was installed in August 2021 and is operated by the von Karman Institute for Fluid Dynamics (Glabeke et al., 2023). In this study, we utilize data from the campaign between 4 August 2021 and 18 July 2022. The lidar measures at 11 different heights, i.e., 34.5, 44.5, 62.5, 79.5, 104.5, 124.5, 149.5, 174.5, 224.5, 274.5, and 324.5 mTAW. Measurement heights are in meter above the Tweede Algemene Waterpassing (mTAW), implying that the average sea level at low tide in Ostend (Belgium) is used as the zero level. This value lies 2.3 m below mean sea level.
  - EPL and LEG: Northeast of the cluster, two more lidars are installed on the Europlatform (EPL) and Lichteiland Goeree (LEG) platforms, respectively. Data is provided by the Wind Energy Research Group of TNO, who operates the lidars (Verhoef et al., 2020; Bergman et al., 2022). LEG features a Leosphere WindCube lidar v2.1 providing measurements at eight heights, i.e., 62, 90, 115, 140, 165, 190, 215 and 240 m above mean sea level. EPL is equipped with a ZX 300M lidar (Verhoef et al., 2020), with data provided at 63, 91, 116, 141, 166, 191, 216, 241, 266, and 291 m above mean sea level.
  - BSB: Inside the cluster, data from the ZX 300M lidar installed at the TenneT platform Borssele B (BSB) and operated by the KNMI is used (Knoop and de Jong, 2023), with measurements at 11 heights, i.e., 59, 83, 108, 117, 139, 164, 190, 201, 224, 244, and 294 m above mean sea level.
- 170 Considering the prevalence of southwesterly and, to a lesser extent, northeasterly wind directions at this site, the combination of WHi, BSB, EPL, and LEG provide a suitable alignment to include a freestream, intra-cluster wake, and far wake observation for these wind directions. All lidar observations come in the form of 10-minute averages. Both lidar measurements and model output are interpolated to a reference hub height of 107 m height above ground level, corresponding to the hub height of many wind turbines within the domain of interest.

**Table 3.** Description of the lidars used in this study. Data availability is expressed as the percentage of time each lidar was operational relative to the full simulation period (2021–2023). Timestamps of missing data are also accounted for in the table.

|                          |                                                   |                                       | Data availability (%) |           |
|--------------------------|---------------------------------------------------|---------------------------------------|-----------------------|-----------|
| Lidar                    | Location                                          | Operational period in WRF time window | Wind                  | Wind      |
|                          |                                                   |                                       | Speed                 | Direction |
| Westhinder (WHi)         | 51°23′18.74″N, 2°26′16.18″E                       | August 2021 - July 2022               | 24.8                  | 23.8      |
| Borssele B (BSB)         | $51^{\circ}43'34.90''$ N, $2^{\circ}57'56.09''$ E | January 2021 - September 2023         | 77.5                  | 80.4      |
| Europlatform (EPL)       | $51^{\circ}59'52.51''N$ , $3^{\circ}16'29.32''E$  | January 2021 - December 2023          | 88.1                  | 90.4      |
| Lichteiland Goeree (LEG) | 51°55′60.00″N, 3°40′00.00″E                       | January 2021 - December 2023          | 95.9                  | 92.2      |

Figure 2. Locations of domain of interest. Black dots indicate turbine locations included in the WRF simulations.

# 2.3.2 SAR Data



The SAR data used in this study are retrieved by the Sentinel-1 European Space Agency (ESA) mission. More specifically, there are two satellites—Sentinel-1A (2014–present) and Sentinel-1B (2016–2021)—equipped with C-band Synthetic Aperture Radar (SAR) instruments operating at 5.405 GHz. Both satellites follow a sun-synchronous polar orbit at an altitude of 693 km, with a repeat cycle of six days (Hasager et al., 2024). Over the Southern Bight of the North Sea, the analyzed time instances occur approximately at 06:00 UTC and 17:40 UTC, respectively. SAR provides wind speed values at 10 meter height and the pixel has a resolution of 500 meters. The wind speed is calculated as a reflected signal, therefore it depends on the surface roughness variations and the terrain characteristics. The data are accessed through the Danish Technical University (DTU) Wind Energy Department (https://science.globalwindatlas.info/#/map/satwinds last access: 30 July 2025).

https://doi.org/10.5194/wes-2025-202 Preprint. Discussion started: 17 October 2025

© Author(s) 2025. CC BY 4.0 License.




# 2.3.3 Data Filtering

Two different types of filtering are used for the lidar measurements and the SAR data, respectively. For the lidars, we apply a threshold and discard wind speed values outside the range of [0.5,70] ms<sup>-1</sup>. The lower bound was set to discard very low wind speeds in which the wind direction becomes ill-defined. A similar approach was taken by Pentikäinen et al. (2023). It was verified that this bound does not significantly impact wind speed validation results. The upper bound discards spuriously high measurements. The resulting interval retains relevant wind speed values for wind energy purposes, including wind speeds below cut-in and above cut-out. This filtering results in an additional data removal of approximately of 1-2% of the total values presented in Table 3, depending on the location. In parallel, the same filtering is applied to the WRF wind speeds in both the NWF and WF simulations (discarding approximately 0.3% of the WRF timestamps). We then use the common timestamps of both filtered lidar and WRF datasets for evaluating both wind speed and wind direction.

The SAR data are polluted by wind turbines and ships in the area of interest, creating local outliers in the retrieved wind fields. In this case, we use a Hampel filter to detect the outlier values (Hampel, 1971). This method detects localized outliers of a series of data  $x = \{x_1, x_2, ..., x_n\}$  using the median absolute deviation (MAD) in a sliding window of size  $W_i$ . The median value  $m_i$  is computed across the sliding window and the MAD is defined as:  $\text{MAD}_i = \text{median}(|x_j - m_i|)$ , where  $x_j$  are the values in the sliding window  $W_i$ . The MAD is scaled to be comparable to the standard deviation for Gaussian data as:  $\sigma_i = 1.4826 \cdot \text{MAD}_i$ . A point  $x_i$  is considered as an outlier if  $|x_i - m_i| > n_\sigma \cdot \sigma_i$ . In this study, to detect the outliers, we use a sliding window  $W_i = 3$  and set  $n_\sigma = 3$  to prevent excessive outlier detection and ensure robustness without over-filtering the data.

#### 2.3.4 Evaluation metrics

Different evaluation metrics are used to provide a comprehensive understanding of model performance. More specifically, the bias, the Centered Root Mean Squared Error (cRMSE), the Pearson correlation coefficient (r), and the Earth Mover's Distance (EMD) are computed for the wind speed and the wind direction. The mathematical definitions of these metrics are provided in Appendix A1. The same metrics have been employed in several studies to evaluate model performance in calculating wind fields (Rosencrans et al., 2024; Olsen et al., 2025).

The bias score measures the systematic error between the simulations and the observations, indicating a consistent overestimation (positive bias) or underestimation (negative bias). The cRMSE is used to quantify the average error magnitude between the model and the lidars, emphasizing the variability around the mean value. The EMD evaluates the similarity between two distributions by calculating the minimum cost required to transform one distribution into another, thereby facilitating the comparison of the overall shape and distribution of predicted and observed wind speed and direction data. The Pearson correlation coefficient (r) measures the linear relationship between predicted and observed values. While bias, cRMSE, and EMD values closest to zero indicate better model performance, the Pearson correlation coefficient (r) ranges from -1 to 1, with values closer to 1 signifying a stronger direct linear relationship between the model and the lidar measurements.

https://doi.org/10.5194/wes-2025-202 Preprint. Discussion started: 17 October 2025

© Author(s) 2025. CC BY 4.0 License.

To compute means on the wind direction, the circular means are applied (Mardia and Jupp, 2009). The cRMSE equation for circular variables (A6) has been adopted from the study of Rosencrans et al. 2024. All metrics are computed using the built-in functions of the NumPy (Harris et al., 2020) and SciPy (Virtanen et al., 2020) Python packages. Furthermore, the wasserstein\_circle function by the Python Optimal Transport (Flamary et al., 2021) package is used to calculate the EMD for the wind direction.

#### 3 Results





This section discusses the simulation results and compares them to observations. First, Section 3.1 investigates the atmospheric conditions during the simulation period, focusing on dominant stability regimes and wind roses. Next, Section 3.2 compares model results to lidar observations. In addition to the validation of the full three-year dataset, the impact of stability on model performance is quantified. Further, wind fields along the lidar transect (see Figure 2) are analyzed to focus more specifically on wake deficits (Section 3.2.3) and blockage (Section 3.2.4). Finally, Section 3.3 presents a comparison of simulation results to SAR observations.

#### 3.1 Atmospheric Conditions

Two main atmospheric characteristics are considered for the investigation, the MO stability (Equation (4), and Table 2) and the wind fields presented as wind roses, assessed in the four lidar locations. Figure 3 illustrates the monthly-averaged MO distributions for the years 2021 (a), 2022 (b), and 2023 (c). The MO values are calculated using WRF at the BSB lidar probe both in the NWF (left bars indicated with 'o') and WF (right bars indicated with 'x') simulations. Despite the intra-annual variability in ABL stratification, the monthly MO distributions indicate that very unstable conditions prevail over the three year-long analysis. Peaks of the very unstable atmospheric conditions predominantly occur in January, as well as during late summer and autumn.

Stable and very stable atmospheric conditions are primarily observed from late winter through early summer, while their occurrence is significantly reduced during autumn. Specifically, June 2021, May 2022, and June 2023 exhibit notably high frequencies of stable and very stable atmospheric conditions. Furthermore, a significant number of neutral cases is also observed. The frequency of neutral or near-neutral conditions is associated with strong winds, which enhances the importance of shear effects over buoyancy. For example, in February 2022, severe storms occurred (Ivanova et al., 2025), resulting in a higher frequency of neutral events over the Southern Bight of the North Sea.

Overall, the predominance of unstable to very unstable events has also been highlighted in various studies in the last decade. Archer et al. (2016) demonstrated that the marine ABL is mostly unstable in the U.S. northeastern coast during a 2003-2011 measurements campaign. Similar conditions have been mentioned in the work of Rosencrans et al. (2024) analyzing the ABL stratification on the Rhode Island–Massachusetts area in a WRF execution from September 2019 to September 2020. However, it is important to highlight that the intra-annual MO length distributions differ, as ABL stability conditions are highly dependent

**Figure 3.** ABL classification based on WRF-estimated MO length at BSB location over the three-year simulation period. The stability classes are described in Table 2. On the y-axis, the frequency of each ABL condition is depicted, and on the x-axis, the months of each year, from 1 (January) to 12 (December). Bars marked with 'o' and 'x' correspond to NWF and WF cases, respectively.

on the area of interest. Last, focusing on the North Sea, analogous conclusions have been reached in the North Sea using WRF simulations in recent works (Porchetta et al., 2024; Palatos-Plexidas et al., 2024).

In addition to the intra-annual stability distributions in the BSB location, the influence of the wind turbines on the ABL stratification should be addressed. The inclusion of wind turbines affects the intra-farm ABL stratification, resulting in an increase of 5 to 15% of the very unstable events during the three-year period. In addition, an increase in very stable instances is observed, while neutral and close-to-neutral cases are slightly reduced. The increase in very unstable and very stable cases may be attributed to the wind farm-induced wake, which leads to lower wind speeds and, consequently, reduced friction velocity magnitudes  $u^*$ .

Figure 4 illustrates the wind roses from the WF simulations (top), as well as the lidar-derived wind roses (bottom) at the analyzed locations. The investigation over the three-year period indicates that the predominant wind direction at all locations is consistently from the southwest. At the WHi location, WRF provides a slight shift to westerly wind direction compared to the lidar observations. A similar trend is noticed in the cases of LEG and EPL locations, where events between southwest to west wind directions are more prevalent in the model. Similar behavior in WRF has been reported in the work of Kalverla et al. (2019), where winds were 10° clockwise-biased. At the BSB probe, the wind rose trends align between the model and the measurements, offering a highly accurate comparison with the lidar data.

**Figure 4.** Top: Wind roses calculated at WHi (A), BSB (B), EPL (C), and LEG(D) probe locations using the WRF model including the WFP scheme in the model. Bottom: Wind roses calculated at WHi (E), BSB (F), EPL (G), and LEG(H) using the lidar measurements. Both the lidar measurements and the model output are interpolated at the hub height.

#### 3.2 Model Validation with lidar observations

# 3.2.1 Full Three-Years Analysis



Figure 5 depicts wind speed metrics calculated for the WRF simulations against the lidar measurements over the full three-year-long period. Furthermore, the ERA5 data are extracted from the nearest grid point and interpolated to the hub height, serving as a baseline. Specifically, the bias score in Figure 5 (A) highlights that the NWF simulation as well as the ERA5 have a negative bias for all locations other than BSB. Since the WRF model is initialized with ERA5 data, this negative bias may propagate into the WRF simulations. The wind speed reduction imposed by the Fitch WFP leads to a slightly worse bias for the WF simulations than the NWF runs. In both NWF and WF cases, WRF demonstrates enhanced consistency with the observations, while ERA5 exhibits greater deviations. At the BSB location, model-estimated wind speeds show a unique positive bias in both the NWF scenario and the comparison with ERA5 data. Taking into account that the prevailing wind direction during the analysis period is southwesterly, the near-wake produced by the Belgian cluster and assessed at the BSB location is more





**Figure 5.** Evaluation metrics for wind speeds against lidar data, interpolated to hub-height at the four lidar locations: Bias (A), cRMSE (B), EMD (C), and r (D) scores. Darker colors signify poorer alignment with the observations.

accurately represented by the WF simulations compared to the NWF runs, resulting in a lower and negative bias. In contrast to the bias score, the cRMSE of ERA5 indicates a slightly more accurate estimation of wind speed compared to the NWF and WF simulations. However, among the two WRF cases, WF configuration outperforms the NWF runs. The EMD scores (Figure 5 (C)) shows that WF achieves the best fit against BSB observations, while exhibiting slightly weaker agreement with the other lidars compared to the NWF simulations. Last, in Figure 5 (D), the Pearson correlation shows that the wind speeds among WF, NWF, and ERA5 are highly correlated with the lidars observations, with a minimum r value equal to 0.90 for the WRF model, and 0.93 for the ERA5 data. In summary, a consistent negative bias is observed in both WRF and ERA5 datasets across all locations, except for BSB. This bias is likely driven by the inherent underestimation in the ERA5 data, which serve as input for the WRF simulations. Consequently, aggregated over the full three-year simulation period, wind speed is slightly underestimated within the analysis domain in both NWF and WF simulations.

Figure 6 presents bias (A), cRMSE (B), EMD (C), and circular correlation  $\rho$  (D) for wind directions, validating the WF, NWF, and ERA5 cases against the lidar observations. Wind speed and wind direction evaluations vary across locations, with wind direction showing greater variability overall. First, it is important to highlight that the WF case yields more accurate wind direction estimates than the NWF simulations across all locations. Focusing on the BSB location, the wind speed metrics show a reduction of about 36% in bias and EMD, while an improvement of 4% in cRMSE is also observed in the WF simulations. Furthermore, when WFP is activated, wind direction bias at BSB is reduced by 65%, with improvements of 8% and 15% in cRMSE and EMD, respectively. However, ERA5 outperforms the WRF model in wind direction validation. The cRMSE, EMD, and  $\rho$  at BSB indicate less consistent wind direction alignment with the lidars compared to the other locations. The increased discrepancies between WRF and lidar measurements compared to ERA5 data can be attributed to a few reasons. The coarse resolution of ERA5 can effectively average out spatial variability and reduce noise. Furthermore, the PBL physics schemes that are used in WRF may introduce uncertainty in near-surface turbulent mixing. Specifically, slightly higher cRMSE values in WRF, compared to ERA5 have been observed in the literature (Pronk et al., 2022). In that work, the authors emphasize that


**Figure 6.** Evaluation metrics for wind directions against lidar data, interpolated to hub-height at the four lidar locations: Bias (A), cRMSE (B), EMD (C), and r (D) scores. Darker colors signify poorer alignment with the observations.

ERA5 can outperform downscaled WRF simulations in certain cases. Additionally, they report a consistent negative bias in wind speed.

#### 3.2.2 Impact of ABL Stability on Model Performance

ABL stability conditions have been shown to play a significant role in influencing wind shear, buoyancy, and turbulence production, which in turn affect the width and length of downwind wind farm wakes (Rosencrans et al., 2024; Palatos-Plexidas et al., 2024). Therefore, it is essential to quantify these effects and assess the performance of WRF under varying ABL stratification. Figure 7 depicts the wind speed scores estimated at the hub height. The ABL stability classes are determined at the BSB location using the MO length derived from the NWF simulations. The black dashed lines at each plot characterizes the ideal score values, while the colored solid and dashed lines represent the NWF and WF simulations at each location, respectively. EPL and LEG locations exhibit comparable patterns, with slightly greater wind speed underestimation at LEG during weakly stable to very stable conditions. In contrast, the wind speed assessed at BSB shows larger discrepancies. In addition, at this location the influence of wind turbines in the WF simulation is more evident where a negative bias due to the wake is always observed (wind speed is underestimated). At WHi, the WF case provides a very accurate match and especially in neutral, and stable cases where the bias is close to zero. Similar trends are also indicated in the EMD scores, where reduced distribution-matching appears at BSB, and closer match to observations is noticed at WHi, EPL, and LEG. Across very unstable to neutral conditions, activating the WFP scheme in the model leads to a marked improvement in accuracy, especially in regions that are mostly affected by wind farm wakes, as evidenced by reductions in bias and EMD scores.

The cRMSE is around  $2 \text{ ms}^{-1}$  for all the cases, while there are subtle differences across the domain under different ABL conditions. Specifically, during very unstable to close-to-neutral conditions WHi, BSB, and EPL generate a reduced cRMSE, while LEG provides the best cRMSE during stable-neutral conditions. Last, correlation r is higher than 0.80 when stability conditions are near-neutral and stable, while during more extreme ABL stratification, correlation values are lower and vary from 0.7 to 0.8. The increase in cRMSE under very stable conditions is also reflected in the Pearson correlation, which decreases



**Figure 7.** Comparison of WRF-assessed hub-height wind speed values against lidars: Bias (A), cRMSE (B), EMD (C), and *r* (D) scores. ABL classes are evaluated at BSB using NWF simulations. Solid lines represent NWF runs, while dashed lines correspond to WF simulations. The distance from the black dashed line indicates the degree of misalignment with observations.

when transitioning from stable-neutral to very stable regimes. These findings align with the correlation analysis between WRF and sonic anemometer measurements in the Horns Rev wind farm (Peña and Hahmann, 2012), which shows that model uncertainty increases as atmospheric stability deviates from neutral, i.e., under very stable or very unstable conditions.

Wind direction scores presented in Figure 8 indicate different trends compared to the wind speed metrics. Focusing on the bias score, the best alignment is observed at BSB, and specifically when the ABL conditions vary from very unstable to stable-neutral. The cRMSE reaches its highest values during periods of extreme ABL stratification—both very stable and very unstable conditions—across all locations, with particularly pronounced misalignment observed at BSB and WHi during very unstable events. Furthermore, cRMSE underscores the difficulty in accurately capturing wind direction—particularly at the BSB location under unstable-neutral stratification. At BSB, EMD suggests that during very unstable to unstable-neutral conditions, there is a discrepancy of around  $7^{\circ}$  to  $10^{\circ}$  in the wind direction distributions, which reduces when transitioning to neutral and stable stratification. On the other hand, higher discrepancies occur during weakly stable to very stable conditions at EPL and WHi locations. In the case of wind direction, circular correlation  $\rho$  provides comparable results to cRMSE, where the model performs less accurately during very unstable conditions, and the best matching is when ABL stratification is nearneutral. More precisely, during very unstable events  $\rho$  drops down to approximately 0.63 and 0.61 at WHi and BSB locations, respectively, indicating limited consistency in directional variability during these conditions. The high cRMSE, and the low



**Figure 8.** Comparison of WRF-assessed hub-height wind direction values against lidars: Bias (A), cRMSE (B), EMD (C), and *r* (D) scores. ABL classes are evaluated at BSB using NWF simulations. Solid lines represent NWF runs, while dashed lines correspond to WF simulations. The distance from the black dashed line indicates the degree of misalignment with observations.

circular correlation observed at BSB, and WHi locations during very unstable events may be attributed to lower winds and enhanced turbulence. Especially at BSB there is an important influence of the wind turbines in the flow, impacting the overall cRMSE and  $\rho$ . The applied cut-in threshold in wind speed (see Section 2.3.3), results in a significant improvement on the wind direction cRMSE and  $\rho$  scores during very unstable events (Figure A2), while wind speed metrics remain unaffected (Figure A1).

The validation analysis under different stability conditions reveals differences in model performance across the various locations. We should highlight that overall the WF simulation performs better across the locations. However, in addition to the analysis aggregated over the full three-year simulation period presented here, specific interest also lies in analyzing wind speeds across different magnitudes and specific wind directions to focus more on wake effects, as discussed in the next section.

# 3.2.3 Assessment of Wind Speeds Along the Transect

In this section, we describe the influence of the wind speed magnitude on the intra-cluster and downstream wind speed deficit by comparing both the NWF and WF simulations with lidars across the transect that connects WHi, BSB, and EPL locations, with an almost perfect collinear alignment of these lidars (see Figure 2). The analysis is based on timestamps where lidar measurements are commonly available, representing about 21.7% of the total time instances. The limitation is primarily due



**Figure 9.** Wind speed classes frequency when wind direction is SW (11.9%) or NE (6.9%), respectively. The frequency is based on the common timestamps of the lidars and not over the full period of analysis.

to the availability of WHi measurements (see Table 3). The study focuses on two predominant wind direction categories: southwest (SW) and northeast (NE), and the frequency of each wind speed class is depicted in Figure 9. This configuration allows both SW and NE wind directions to effectively characterize the wind fields upstream and downstream the Belgian-Dutch cluster. Both wind direction and wind speed classification is based on the average values across the three locations.

Figure 10 depicts wind speed along the transect connecting WHi, BSB, and EPL under the four different wind speed classes and SW wind direction. The distributions at each location are clipped at the 1st and 99th percentiles solely for illustrative purposes. Overall, the WF simulations slightly underestimate the wind speed, and this is corroborated by the bias scores in the full-period-heatmaps presented in Figure 5 (A). More precisely, under SW winds, the model underestimates wind speeds in the [3, 5) ms<sup>-1</sup> range, particularly at WHi and EPL, with discrepancies exceeding 1 ms<sup>-1</sup>. In contrast, wind speeds in the [5, 8) and [8, 11) ms<sup>-1</sup> ranges show strong agreement, although a slight underestimation, around 1 ms<sup>-1</sup> persists at BSB when wind speed varies between 5 to 8 ms<sup>-1</sup>. At higher wind speeds ([11, 15) ms<sup>-1</sup>), the model performs well at WHi and EPL but there is limited accuracy at BSB, suggesting intensified wake effects within the wind farm. Wake formation under SW winds is mostly observed in the dense zone of the Belgian wind farms and remains relatively constant downstream the Borssele wind farm.

In Figure 11, under northeasterly winds, the model underestimates wind speeds in both the [3, 5) and [5, 8)  $\mathrm{ms}^{-1}$  ranges across all locations, while performance improves markedly when wind speed varies from 8 to  $15~\mathrm{ms}^{-1}$ , with a modest underestimation of about  $0.5~\mathrm{ms}^{-1}$  at BSB. Notably, intra-cluster wake behavior differs under NE winds, with a continuous reduction in wind speed observed throughout the wind farm, beginning in the Borssele zone and intensifying through the denser Belgian


**Figure 10.** WRF-assessed Hub-Height Wind speed validated across the WHi, BSB, and EPL transect. Wind direction is SW and it is assessed as the average across the three lidar locations. The vertical lines indicate the Belgian-Dutch cluster (black) and the Belgian-only (red) wind farm areas. The wind speed range on the y-axis varies across classes, aiming to highlight the distribution specific to each class.

wind farms. Furthermore, the wake recovery rate is less smooth at high wind speed magnitudes. These findings highlight the directional dependence of wake dynamics and underscore the need for improved model representation of low-speed flows and intra-cluster wake effects. The wind speed deficits over the Southern Bight of the North Sea are illustrated in Figure 12. Following the transect plots in Figures 10 and 11, greater wind speed magnitudes are associated with more pronounced downstream wind speed deficits.

Figure 13 presents the frequency of the ABL stability conditions at each wind direction case and under the different wind speed range. Under low wind speed conditions, very unstable and unstable events tend to dominate. Conversely, a transition toward near-neutral ABL stratification under high wind speeds is evident in Figure 13. Previous studies (Rosencrans et al., 2024; Palatos-Plexidas et al., 2024; Porchetta et al., 2024) have demonstrated the dependence of wake propagation and wind speed losses on ABL stability, showing that transitioning from neutral to stable and very stable stratification, wakes tend to extend further downstream of wind farms. Similarly, we underscore the relation between wind speed magnitude, ABL stratification, and wake propagation, where during high wind speeds, the frequency of neutral events is enhanced and wakes propagate a few tens of kilometers downstream the Belgian-Dutch cluster.


**Figure 11.** WRF-assessed Hub-Height Wind speed validated across the WHi, BSB, and EPL transect. Wind direction is NE and it is assessed as the average across the three lidar locations. The vertical lines indicate the Belgian-Dutch cluster (black) and the Belgian-only (red) wind farm areas. The wind speed range on the y-axis varies across classes, aiming to highlight the distribution specific to each class.

# 3.2.4 Blockage effects in the Belgian-Dutch cluster

The analysis presented in the previous section, based on SW and NE wind directions, supports the investigation of blockage effects within the Belgian-Dutch wind farm cluster. A reduction in wind speed upstream of the wind farms is evident both in the transects (Figures 10 and 11) and in the wake maps shown in Figure 12. In the case of SW winds (Figure 10), a noticeable wind speed deficit begins approximately 8 km upstream of the wind farm. This reduction is present across different wind speed magnitudes but becomes more pronounced and rapid around 5 km upstream, particularly when wind speeds exceed 5 ms<sup>-1</sup>. For NE wind directions (Figure 11), the blockage effect is also visible, though it is confined to a smaller region—approximately 3 km upstream of the Borssele wind farm, and occurs when wind speeds range between 5 and 15 ms<sup>-1</sup>.

The full wake maps in Figure 12 further confirm the presence of blockage effects under both SW and NE wind conditions. Notably, when wind speeds exceed  $5~\mathrm{ms^{-1}}$ , the upstream wind speed deficit extends over a broad area. Additionally, the geometric configuration of the cluster leads to distinct blockage zones depending on the wind direction. Speedup zones (i.e., negative wind speed deficits) are also observed between the Belgian-Dutch cluster and the UK wind farms.


Figure 12. Wind speed deficits calculated at the hub height, for four wind speed classes. The red dashed line indicates the WHi, BSB, and EPL transect. Subscript 1 indicates that wind direction is SW (top), and subscript 2 presents the NE wind direction cases (bottom). Wind speed deficits are averaged across the occurrences that describe each case.

#### 390 3.3 Comparison of Wake Events with SAR Data

In this section, we focus on a one-by-one comparison of the WRF model output with SAR data. SAR images are retrieved from Sentinel 1A and 1B satellites as explained in Section 2.3.2, and they provide wind speed fields at 10-meter height. Four discrete timestamps have been selected to analyze the wind speed patterns derived from SAR and WRF data, as well as to validate the model performance across the transect that connects the WHi, BSB, and EPL lidars (see Figure 2). These timestamps were selected manually based on a visual appearance of wind farm wakes in the SAR images. Therefore, an additional event from September 2020, outside the defined three-year simulation period, has been deliberately included for analysis. The exact datetimes and the ABL stratification conditions are described in Table 4.

Figure 14 presents the comparison of SAR and WRF wind fields at 10-meter height. The left column shows the SAR-derived wind speeds, denoted by the subscript 1. The middle column depicts the WF wind speeds, indicated by a subscript 2, and in the right column the reference NWF wind speeds are illustrated, and denoted with subscript 3, as a reference case. In case A, higher wind speeds are also related to stable-neutral conditions and a shallow boundary layer (see Table 4). The wake structure of the Belgian-Dutch wind farm cluster is well captured by the WF simulation. However, a slight deviation in wind direction is observed in WRF, exhibiting a positive bias toward more westerly flow. Similar bias has also been highlighted in the wind roses (Figure 4) and in the wind direction validation heatmap in Figure 6 (A). In case B the wake is weaker, but still detectable

**Figure 13.** ABL stability frequency (%) for each wind speed class, based on the two analyzed wind directions, SW (top), and NE (bottom). Lighter colors suggest higher frequency of the stability class.

**Table 4.** Description of the atmospheric conditions on the SAR-WRF comparison timestamps. MOL and planetary boundary layer height (PBLH) are estimated in the NWF simulations.

| Datetime            | WHi        |                 |                      | BSB        |             |                      | EPL        |                 |                      |
|---------------------|------------|-----------------|----------------------|------------|-------------|----------------------|------------|-----------------|----------------------|
|                     | MOL<br>(m) | <b>PBLH</b> (m) | Stability            | MOL<br>(m) | PBLH<br>(m) | Stability            | MOL<br>(m) | <b>PBLH</b> (m) | Stability            |
| 2020-09-19 17:40:00 | 259.5      | 291.9           | Stable-<br>neutral   | 355.0      | 264.6       | Stable-<br>neutral   | 1000       | 326.2           | Neutral              |
| 2021-10-08 17:40:00 | -38.7      | 688.8           | Very<br>unstable     | -27.2      | 729.9       | Very<br>unstable     | -29.9      | 715.9           | Very<br>unstable     |
| 2022-03-25 17:40:00 | -450.9     | 297.6           | Unstable-<br>neutral | -296.1     | 317.1       | Unstable-<br>neutral | -410.8     | 317.6           | Unstable-<br>neutral |
| 2023-08-20 06:10:00 | -51.1      | 249.3           | Very<br>unstable     | -98.8      | 338.9       | Very<br>unstable     | -61.3      | 357.7           | Very<br>unstable     |

https://doi.org/10.5194/wes-2025-202 Preprint. Discussion started: 17 October 2025 © Author(s) 2025. CC BY 4.0 License.





both in SAR image and in the WF simulation. In this case, stability conditions are very unstable and the PBLH is around 700 meters, explaining the attenuated wake pattern. Case C is characterized by higher wind speeds, MO length varies from -300 to -450 m and PBLH is approximately 300 m across the three locations, resulting in a more unstable-neutral stratification profile. Last, in case D, very unstable ABL conditions prevail and the PBLH varies between 250 meters upstream at the WHi location to 358 meters downstream at the EPL area. This is the only SAR image with clear wakes for SW wind conditions. Overall, the patterns of the wakes are accurately represented by the WF simulations, highlighting the importance of using the Fitch WFP scheme in WRF.

In addition to the wind speed deficit maps, Figure 15 presents the transects that connect WHi, BSB, and EPL. The green line illustrates the SAR 10-meter height wind speeds, while the blue and orange lines correspond to the WF and NWF simulation outputs, respectively. Although a Hampel filter (see Section 2.3.3) has been applied to the SAR data, we can still observe a few outlier values, especially in the intra-farm area. Therefore, to further evaluate the performance of the WRF model, hub height-lidar measurements are incorporated when available, depicted as scatter points in Figure 15.

Case A shows good agreement in the representation of the transect wake. Specifically, while the WF simulation tends to underestimate wind speeds upstream and within the Borssele zone, it accurately captures the wind speed deficit in the Belgian wind farms and downstream of the wind farm cluster. Similarly, in Case B, both the intra-farm and downstream wind speed patterns are well represented by the WF simulation, whereas the NWF case fails to capture the wind speed deficit. Although the overall trend is accurately reproduced, the WF simulation tends to overestimate the wind speeds downstream the wind farms. Additionally, a flow acceleration is observed as the wind exits the Belgian-Dutch cluster. This can be attributed to a misalignment in wind direction, with the simulated flow being closer to westerly (see Figure 14, B), causing the transect points to shift toward the southern edge of the wake and partially escape the waked region. Both the lidar measurements and the WF simulation effectively capture the intra-farm wind speed deficit and the downstream flow acceleration. A maximum positive bias of approximately 1 (ms<sup>-1</sup>) is observed at EPL. Although the averaged stability trends shown in Figure 7 (A) appear to contradict this positive bias, it is important to note that Case B represents a discrete timestamp and therefore cannot be considered representative of the average ABL conditions.

Case C also features NE incoming wind conditions. Similar to the previous cases, the WF simulation successfully captures the wind speed deficit trends both within the intra-cluster zone and in the downstream wake recovery region. The WF simulation shows noticeable discrepancies compared to lidar observations, particularly at BSB and EPL, where negative biases of approximately 1.7 (ms<sup>-1</sup>), and 3 (ms<sup>-1</sup>), respectively are observed. MO length is relatively large and negative, indicating unstable to near-neutral conditions. This may be attributed to elevated shear, which can contribute to increased deviations from the lidar measurements. In contrast, at BSB, the NWF simulation exhibits a significantly lower but positive bias. However, in this case, it produces an unrealistic and nearly linear increase in wind speed of approximately 0.5 (ms<sup>-1</sup>) between EPL and WHi. Case D is the only selected timestamp with a SW wind direction. Although the SAR data exhibit considerable noise, particularly within the intra-farm region, the WF simulation is still able to reproduce the wind speed deficit and the downstream recovery trends. In this case, only EPL lidar measurements are available, which show good agreement with the WF simulation.

**Figure 14.** 10-meter height wind speeds: SAR images (left, subscript 1), WF simulations (middle, subscript 2), and NWF simulations (right, subscript 3). Four discrete timestamps are compared where the wind farm-produced wake is present in both cases. The red line depicts the transect among WHi, BSB, and EPL locations.

**Figure 15.** Wind speeds along WHi, BSB, and EPL transect. Comparison between WF (purple), NWF (orange), and SAR (green) at 10 meters height. Validation against WHi, BSB, and EPL lidars (scatter points, when available) at 107 meters height. Similar to Figures 10 and 11, the vertical black and red dashed lines indicate the Belgian-Dutch cluster and the Belgian zone only, respectively.

https://doi.org/10.5194/wes-2025-202 Preprint. Discussion started: 17 October 2025 © Author(s) 2025. CC BY 4.0 License.






#### 4 Conclusions

This study incorporates high-resolution mesoscale simulations using WRF coupled with the Fitch WFP scheme over a three-year-long period (2021-2023) focusing on the Southern Bight of the North Sea. The model is systematically validated using both lidar measurements and synthetic aperture radar (SAR) imagery. More precisely, the performance of the model is evaluated over the three-year simulation period, while additional analysis under varying ABL conditions and wind speed magnitudes is presented. Furthermore, the model-simulated wind speeds are compared against 10-meter wind speeds derived from SAR data.

A key contribution lies in the detailed investigation of upstream, intra-farm, and downstream wake structures, utilizing both WRF model outputs and observational data at selected locations.

Our findings demonstrate that the use of the Fitch WFP, in combination with this particular WRF configuration, is largely able to capture wake effects and improve the general accuracy over a simulation without wind farms included. First, focusing on the time-averaged wind speeds over the full analysis period, the WF simulations provide lower bias values (about 36% reduction) across the domain, along with reduced cRMSE (improved by 4% approximately), and EMD (a decrease of around 36%) in the intra-cluster region (BSB location). Similarly, wind direction metrics are also improved in the WF simulations, as bias is reduced by approximately 65% at BSB, while cRMSE and EMD are improved by approximately 8% and 15%, respectively. The ABL classification based on the values of MO length indicates that very unstable conditions prevail close to the Belgian-Dutch cluster. In addition, this work highlights that when the stability conditions are near-neutral, WF simulations provide an adequate comparison against the lidars, while during extreme stratification the uncertainty of the model increases. Classifying wake events based on wind speed magnitude reveals that, as wind speed increases, atmospheric stability transitions from very unstable to neutral and stable conditions. The WF simulations accurately capture the wake structure when wind speeds range between 5 to 11 ms<sup>-1</sup>. Although the model tends to overestimate intra-farm wind speed deficits at BSB under higher wind speed conditions, it accurately captures the long-distance downwind wakes at EPL (SW wind direction) and WHi (NE wind direction), yielding smaller discrepancies when compared to lidar measurements. The analysis of SW and NE wind directions also provides insights into the development of blockage effects in the Belgian-Dutch cluster. Specifically, an upstream wind speed reduction zone of about 5 km is observed when the wind is SW, while in the case of NE wind direction, this zone can be shorter, around 3 km.

Furthermore, a comparison with SAR images at specific events shows that WF performs well in representing the wake structure generated by the Belgian-Dutch cluster. Both in the intra-farm and the wake recovery regions, WF provides a realistic wind speed deficit. This work can accurately evaluate the model performance by utilizing upwind, intra-farm, and downwind lidar measurements as well as SAR images at specific events. Systematic biases in both wind speed and direction are evident, particularly in the wind speed metrics. However, it should be highlighted that the model is driven by ERA5 hourly data, which consistently introduce a negative bias, affecting the boundary conditions in WRF and propagating wind speed underestimations to the analyzed locations. In addition, due to uncertainties introduced by the PBL schemes in WRF, exploring alternative physics configurations in the future may be warranted to enhance simulation accuracy. Furthermore, benchmarking different wind farm parameterization schemes for this case study is an interesting area of future research.

https://doi.org/10.5194/wes-2025-202 Preprint. Discussion started: 17 October 2025


This work may serve as a resource for future studies, offering a comprehensive three-year analysis of ABL stratification, wake dynamics, and model validation. The importance of mesoscale models is underscored as they can adequately describe the wake effects at large scales, where high-fidelity simulations are still impractical. In addition, the availability of measurements is essential for ensuring the reliability of numerical simulations and for quantifying associated uncertainties. Finally, further improving our understanding of wind farm wake dynamics will require both advances in wind farm modelling and more extensive measurement campaigns.

Code and data availability. The ERA5 hourly reanalysis data (Hersbach et al., 2020) that have been used in this study are publicly available and they can be downloaded from the Copernicus Climate Data Store at https://doi.org/10.24381/cds.bd0915c6. WRF is an open source model and the version used in this study can be found in Github: https://github.com/wrf-model/WRF/releases/tag/v4.5.2. The lidar profiles provided by TNO (EPL and LEG platforms) are publicly accessible: https://offshorewind-measurements.tno.nl/en/data/. The lidar profiles located at the BSB platform are provided by KNMI: https://dataplatform.knmi.nl/dataset/windlidar-nz-wp-platform-10min-1. Last, WHi lidar profiles remain confidential.

# 485 Appendix A: Validation of the model

### A1 Evaluation metrics

The equations used to calculate the wind speed metrics:

$$\operatorname{Bias}_{V} = \frac{\sum_{i}^{N} \left( V_{\mathrm{WRF},i} - V_{\mathrm{lidar},i} \right)}{N},\tag{A1}$$

$$cRMSE_{V} = \sqrt{\frac{\sum_{i}^{N} \left( \left( V_{WRF,i} - \overline{V_{WRF}} \right) - \left( V_{\text{lidar},i} - \overline{V_{\text{lidar}}} \right) \right)^{2}}{N}},$$
(A2)

$$r_{V} = \frac{\sum_{i}^{N} \left( V_{\text{WRF},i} - \overline{V_{\text{WRF}}} \right) - \left( V_{\text{lidar},i} - \overline{V_{\text{lidar}}} \right)}{\sqrt{\sum_{i}^{N} \left( V_{\text{WRF},i} - \overline{V_{\text{WRF}}} \right)^{2}} \sqrt{\sum_{i}^{N} \left( V_{\text{lidar},i} - \overline{V_{\text{lidar}}} \right)^{2}}}.$$
(A3)

Figure A1. WRF-assessed hub-height wind speed against lidar Locations: Bias (A), cRMSE (B), EMD (C), and r (D) scores. ABL classes are assessed at BSB in the NWF simulations. Distance from the black dashed line reflects the degree of misalignment with observations

The equations used to calculate the wind direction scores are defined in this Section:

$$\operatorname{circmean}(\theta) = \operatorname{atan2}\left(\frac{1}{N} \sum_{i}^{N} \sin(\theta_i), \frac{1}{N} \sum_{i}^{N} \cos(\theta_i)\right), \tag{A4}$$

$$Bias_{\theta} = circmean(\theta_{WRF} - \theta_{lidar}), \tag{A5}$$

$$cRMSE_{\theta} = \sqrt{circmean \left(180^{\circ} - ||\left(\theta_{WRF,i} - \overline{\theta_{WRF}}\right) - \left(\theta_{lidar,i} - \overline{\theta_{lidar}}\right)| - 180^{\circ}|\right)^{2}}, \tag{A6}$$

$$\rho_{\theta} = \frac{\sum_{i}^{N} \sin(\theta_{\text{WRF},i} - \text{circmean}(\theta_{\text{WRF}})) \sin(\theta_{\text{lidar},i} - \text{circmean}(\theta_{\text{lidar}}))}{\sqrt{\sum_{i}^{N} \sin^{2}(\theta_{\text{WRF},i} - \text{circmean}(\theta_{\text{WRF}}))} \sqrt{\sum_{i}^{N} \sin^{2}(\theta_{\text{lidar},i} - \text{circmean}(\theta_{\text{lidar}}))}}.$$
(A7)

#### A2 Model Performance under various ABL Stratification conditions

Figures A1, A2 depict similar results with Figures 7, and 8, respectively. However, in this case we do not filter the wind speed and wind directions from WRF when  $U_{\rm WRF} < 0.5$  and  $U_{\rm WRF} > 70~{\rm ms}^{-1}$ . Compared to wind speed metrics in Figure 7, the application of filtering does not significantly affect the overall metrics. However, Figure 8 shows that cRMSE and  $\rho$  exhibit a significant improvement in wind direction when wind speed is filtered.

**Figure A2.** WRF-assessed hub-height wind direction against lidar Locations: Bias (A), cRMSE (B), EMD (C), and r (D) scores. ABL classes are assessed at BSB in the NWF simulations. Distance from the black dashed line reflects degree of misalignment with observations

Author contributions. APP contributed to the investigation, methodology, software, data curation, writing (original draft), and writing (review and editing). SG contributed to the methodology and writing (review and editing). JvB contributed to funding acquisition, supervision, and writing (review and editing). LDC contributed to methodology, supervision, and writing (review and editing). WM contributed to conceptualization, investigation, methodology, writing (review and editing), supervision, project administration, and funding acquisition.

Competing interests. The authors declare that they do not have any competing interests.

Acknowledgements. This research has received financial support from the Flemish Government through the Agency for Innovation and Entrepreneurship (Vlaams Agentschap Innoveren en Ondernemen, VLAIO), within the framework of Cloud4Wake project. Furthermore, this work received funding from the Federal Public Service Economy of the Belgian Federal Government through the Energy Transition Fund, under the BeFORECAST project. Lesley De Cruz acknowledges support from the Belgian Science Policy Office (BELSPO) through the FED-tWIN programme (Prf-2020-017). The authors would like to acknowledge Gertjan Glabeke for the installation and processing of the Westhinder platform lidar dataset. HPC resources are granted by the Flemish Supercomputer Center (VSC), supported by funding from the Research Foundation – Flanders (FWO) and the Flemish Government.

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
