# Peer review of "Assessing the Accuracy of a Three-Year High-Resolution Mesoscale Wind Farm Wake Simulation with Lidar and Satellite Radar Data"

_Wind Energy Science, 2025_

## Referee Comment (RC1)

**Assessing the Accuracy of a Three-Year High-Resolution Mesoscale Wind Farm Wake Simulation with Lidar and Satellite Radar Data**

The paper employs WRF with the Fitch wind-farm parameterization to simulate three years of operation in the Belgian—Dutch offshore cluster. Results are compared with Lidar data at four sites and with SAR data, classified by stability regime to assess performance under different conditions.

Overall, the paper is clear, well organized, and technically competent. However, the study's novelty is not well articulated. The paper currently reads as a case-study application of an existing parameterization rather than addressing a clear research gap. The introduction lists several limitations of Fitch's parameterization and mentions multiple validation studies, yet the manuscript does not explicitly explain what new insight this paper brings. Clarifying the main research question and the key take-home messages would significantly improve the manuscript.

**General comments**

- Please clarify the specific research question or gap being addressed. Is the purpose to validate Fitch's parameterization, or to evaluate WRF's long-term predictive skill for offshore wakes?
- Several metrics show that the wind-farm (WF) simulations do not outperform the no-wind-farm (NWF) simulations, which seems inconsistent with some of the conclusions.

**Specific comments**

- Line 105: Please specify whether one-way or two-way nesting was used. I assume one-way feedback given the use of nested inner domains (d03).
- Equations (1-3): Fitch's implementation in WRF applies an energy correction to the momentum and TKE tendencies. The equations presented should reflect the actual WRF formulation, not only the original Fitch et al. (2012) expressions.
- Line 182: The statement "The wind speed is calculated as a reflected signal" is unclear. Which wind speed do you mean: surface, 10 m, or another height? Note that the 10 m wind speed derived from SAR is inferred from radar backscatter by assuming neutral stability. So, it is not a measured quantity, but a processed one.
- For completeness, please include a description of the EMD metric in the Appendix.
- Line 256/Eq. A4: Clarify how wind direction is computed. Are wind components rotated from model to Earth coordinates?
- Figure 5: Why does including wind-farm effects increase the bias relative to observations at most sites, given that both WF and NWF simulations share the same background bias from ERA5?
- Figure 5 (cRMSE): ERA5 appears to outperform both WRF setups when compared to centralized error. Please discuss why.
- Model sensitivity: Since the WF simulation rarely performs best (only EMD metric at BSB location), could parameter choices (e.g. TKE coefficient = 1 instead of 0.25) influence the results? A brief sensitivity analysis might help.
- Figure 6: The same issue applies to wind-direction metrics.

- Figure 15: How do you reconcile the comparison with SAR, which assumes neutral stability, while WRF simulates varying stratification? The WRF-Lidar comparison may be more representative, and the SAR transects can be more of a qualitative reference.
- Line 441 and Conclusions: The claim that "the model is systematically validated" and that "WF simulations improve the general accuracy" is not fully supported by the presented results. Please moderate these statements.
- The reported 36% bias reduction appears inconsistent with Fig. 5a, where WF performs worse at most sites. Please clarify how this number was derived.
- The suggestion that future work should "benchmark different parameterization schemes" is valid but not novel as this topic has been addressed many times in the literature, particularly Fitch's parameterization and the Explicit Wind-farm Parameterization (EWP).

**Minor comments**

- Line 6: Spell out "1 km" as "one kilometer".
- Line 118: "...is the number of turbines..."

**Recommendation: Major Revision**

The manuscript presents an interesting long-term dataset comparison and is technically solid, but it currently lacks a clear research question and contains apparent inconsistencies between results and conclusions. A major revision is recommended to better establish the study's novelty, refine WRF's setup if necessary, and improve the interpretation of the results.